# Alloying Iron into Palladium Nanoparticles for an Efficient Catalyst in Acetylene Dicarbonylation

**DOI:** 10.3390/nano12213803

**Published:** 2022-10-28

**Authors:** Yuchen Zhang, Jianhui Zhang, Zongcheng Liu, Yiyi Wu, Yu Lv, Yadian Xie, Huanjiang Wang

**Affiliations:** Key Laboratory of Low-Dimensional Materials and Big Data, School of Chemical Engineering, Guizhou Minzu University, Guiyang 550025, China

**Keywords:** bimetallic catalysts, synergistic effects, acetylene dicarbonylation, metal leaching, theoretical calculations

## Abstract

Motivated by the prominent catalytic performance and durability of nanoalloy catalysts, the Pd-based bimetallic nanoalloy catalysts were prepared using an aqueous reduction method. The Fe-Pd bimetallic nanoalloy catalyst (nano-Fe/Pd) demonstrated 98.4% yield and 99.7% selectivity for the unsaturated 1,4-dicarboxylic acid diesters. Moreover, the inductively coupled plasma (ICP) analysis shows that the Pd leaching of the catalyst can be effectively suppressed by alloying Fe atoms into the Pd crystal lattice for acetylene dicarbonylation. The detailed catalyst structure and morphology characterization demonstrate that introducing Fe into the Pd nanoparticles tunes the electronic–geometrical properties of the catalyst. Theoretical calculations indicate that the electrons of Fe transfer to Pd in the nano-Fe/Pd catalyst, enhancing activation of the C≡C bond in acetylene and weakening CO absorption capacity on catalyst surfaces. Alloying Fe into the Pd nanocatalyst effectively inhibits active metal leaching and improves catalyst activity and stability under high-pressure CO reactions.

## 1. Introduction

Acetylene carbonylation is an excellent atom economy reaction that is independent of petroleum-derived feedstocks. Therefore, using carbon monoxide (CO) as a carbonyl source under the Fe, Co, Ru, Pd, or Rh-based catalysts to yield carbonyl-containing compounds, such as acrylic acid, maleic, or succinic anhydride, has gained increasing attention from academia and industry [1,2,3]. One of these, acetylene dicarbonylation, provides a cost-effective way to synthesize α, β-alkynyl esters, unsymmetrical maleate esters, and their derivatives [4]. Heterogeneous catalysts are one of the promising replacers for homogeneous catalysts despite various supported Pd catalysts being designed, such as Pd_1_/AC ([Pd(CO)I_4_(O=AC)]^2−^) [5], Pd/AC (nanosheet) [6], and Pd/Fe_2_O_3_ [7,8], which have good activity and selectivity. However, it has been reported that supported Pd nano-catalysts are more likely to generate soluble metal carbonyl species and experience severe metal leaching in high-pressure CO reactions [9]. In addition, the catalytic performance of supported Pd catalysts relies mainly on the surface structure and shape of the active metal, as well as the type and micromorphology of the carrier [5,6,7,8].

Bimetallic nanocatalysts are a type of heterogeneous catalyst that contains two metal components within a single catalyst and whose composition, size, and configuration can be reasonably controlled. More importantly, it exhibits higher catalytic activity and better selectivity and stability than those of monometallic analogs due to the synergistic effects between the two metallic elements [10,11,12,13,14,15,16,17]. In recent decades, several laboratories have focused on the exploitation of non-noble metals and Pd nanoalloy catalysts, such as Pd-Co [11], Pd-Ni [12], Pd-Cu [13], Pd-Mn [14], and Pd-Fe [15,16,17]. For example, Yang et al. [16] found that Pd-Fe nanoalloy catalysts have high activity and long-term durability for formic acid oxidation reactions after the Fe atom was introduced. Moreover, Guczi, L. [17] found that alloying Pd catalysts with certain amounts of Fe (Fe/Pd < 1) led to good catalytic performance and durability for the synthesis of methanol from synthesis gas.

Inspired by Pd-Fe nanoalloy catalysts, which behave with brilliant catalytic performance and durability for catalytic reactions with CO, herein, a series of Pd-based bimetallic nanocatalysts were prepared using a simple aqueous-based reduction procedure. It is gratifying that the addition of Fe to the Pd-based nanoalloy catalyst not only improves activity but also stability in acetylene dicarbonylation. Subsequently, electron microscopy and spectroscopy were used to characterize the structure and morphology of the synthesized catalysts. The additional effect of Fe in the nano-Fe/Pd catalyst were revealed using in situ FT-IR and DFT calculations on the Vienna ab initio simulation package (VASP) [18]. Our results indicate that the electrons of Fe naturally transfer to Pd, enhancing electron density around Pd-adjacent Fe species in the nano-Fe/Pd catalyst, which improves the activity and stability of the Pd-based catalysts for acetylene dicarbonylation at high-pressure CO.

## 2. Materials and Methods

### 2.1. Materials

Analytical reagents including FeCl_3_•6H_2_O, NiCl_2_•6H_2_O, polyvinyl pyrrolidone (PVP, average molecular weight of 5800), sodium borohydride (NaBH_4_), potassium iodide (KI), potassium thiosulfate (K_2_S_2_O_3_), methyl benzoate, dimethyl oxalate (DMO), dimethyl maleate (DMM), dimethyl fumarate (DMF), methanol, and ethanol were purchased from Shanghai Macklin Biochemical Co., Ltd., (Shanghai, China). PdCl_2_•2H_2_O, CoCl_2_•6H_2_O, and CuCl_2_•2H_2_O were purchased from Adamas Reagent Co., Ltd., (Shanghai, China). Carbon monoxide (CO), acetylene, air, hydrogen, and nitrogen were purchased from Guizhou Guorui Gas Technology Co., Ltd., (Guiyang, China).

### 2.2. Catalyst Synthesis

The nanoalloy bimetallic catalyst was prepared by an aqueous-based reduction procedure [19]. For example, 1.0 g of PVP was dissolved in 100 mL of PdCl_2_ and FeCl_3_ solutions (0.1 mmol of PdCl_2_ and 0.01 mmol of FeCl_3_). After 0.5 h of stirring in an ice bath at 0 °C, 10 mL of 0.1 M NaBH_4_ aqueous solution was quickly added into the mixture solutions. After another 2 h of stirring and centrifuging at 15,000 rpm for 10 min at room temperature, the target product was obtained by freeze-drying. Analogously, the monometallic nano-Pd catalyst and other bimetal nanoparticles were also prepared from the corresponding metal salts. If a few minor Na and Cl ions remain, those unnecessary species could be removed by dialyzing.

### 2.3. Catalyst Characterization

The phase structure of the catalyst was explored by XRD on a PANalytical X’Pert PRO X-ray diffractometer (PANalytical B.V., Almelo, Holland), with Cu Kα as the radiation (λ = 1.5419 Å), a scanning speed of 5°/min, and a scanning range of 2θ = 10~80°. The micro morphology of the catalysts was observed by a FEI Tecnai G2 F20 TEM (FEI Company, Hillsborough, OR, USA). The metal element content of the catalyst was determined by an Agilent ICP-AES and EDX (Agilent Technologies Co., Ltd., Santa Clara, CA, USA). XPS was recorded on a X-ray photoelectron spectrometer (Thermo Fisher Scientific, Waltham, MA, USA) with an Al Kα X-ray excitation source. The survey spectra were obtained at 100 eV of pass energy and 1.0 eV of step size. Additionally, high-resolution spectra were obtained at 50 eV of pass energy and 0.05 eV of step size, respectively. The C1s peak (284.8 eV) was used to correct the charging effects on all samples. The program XPS Peak 4.0 was used to analyze spectra. All subpeaks were fitted using a linear combination of Gaussian and Lorentzian line shapes, and baselines were generated using an integrated shirley background subtraction model. The H_2_-TPR curve of the samples was monitored on a multifunctional dynamic adsorption instrument equipped with a TCD detector (Micromeritics chemisorb 2920, purchased from Micromeritics, Norcross, GE, USA). Next, 0.08 g of the catalyst samples were placed into a quartz U-tube and heated in Ar flow at 200 °C for 1 h at a rate of 10 °C/min to remove any remaining water. After the sample was cooled, the flow of Ar changed to a 10% H_2_/Ar mixture. Following the stabilization of the baseline, the temperature was turned up to 700 °C at a ramp rate of 10 °C/min to reduce the samples, and the TCD was used to record the hydrogen consumption.

In situ Fourier Transform Infrared Spectroscopy (FT-IR) measurements of CO adsorption at a resolution of 4 cm^−1^ were performed using a Vertex70 (Bruker, Billerica, MA, USA) infrared spectrometer equipped with a mercury cadmium telluride (MCT) detector. Briefly, the reference background signal was collected after the sample was treated with He at 200 °C for 2 h and cooled to room temperature. Subsequently, CO was introduced into the cell and balanced for 30 min. Then, He was used to sweep the dissociative and physically adsorbed species on the catalyst, and CO-IR spectra were recorded at 25 °C at 1, 2, 3, 4, and 5 min.

### 2.4. Computational Methods

The DFT computations were performed by using the Vienna ab initio simulation package (VASP). The Perdew–Burke–Ernzerhof (PBE) functional form of the generalized gradient approach (GGA) was used to describe the exchange-correlation potential. The core electrons are described using projector augmented wave (PAW) potentials. An energy cutoff of 400 eV for the plane-wave basis and a 3 × 3 × 1 Monkhorst-Pack k-point sampling of the Brillouin zone were used in the calculations. Furthermore, 0.05 eV/Å was set as the convergence threshold for the total force acting on the system. The adsorption energies (*E*_ads_) were determined as the energy difference between the adsorbate-containing surface (Total) and the clean surface with the adsorbing molecule in the gas phase (*E*_slab_ + *E*_g_).
(1)$$E_{\mathrm{ads}}=E_{\mathrm{total}}-(E_{\mathrm{slab}}+E_{g})$$

The surfaces of Pd (111) and Fe/Pd (111) are modeled as four layers of a slab with 16 atoms per plane in a 4 × 4 surface unit cell. During the computations, the topmost atomic layers of each slab (with and without adsorbed molecules) were allowed to relax, and the two bottom ones were kept fixed in the bulk position. Meanwhile, a vacuum zone of 15 Å was introduced for the surface slabs along the z-direction to eliminate interaction between the periodically repeated slabs and adsorbates. The data used in the Section 3 are derived from the optimized models.

### 2.5. Catalyst Evaluation

Typically, the methanol (20.0 mL) and catalyst (10 mg) were placed into the 50 mL autoclave (Yanzheng Instrument Ltd., Shanghai, China). After sealing the reactor, it was loaded with 11 mmol of acetylene, a certain amount of CO, and dry air (total pressure of 4.0 Mpa) in sequence. Then, the mixture was stirred (500 rpm) for 15 min, followed by heating to the target temperature and was then kept for several hours. When the reaction was complete, the autoclave was depressurized once it had cooled to room temperature. Before workup, the solid catalyst was separated by centrifugation, and 30 mg of internal standard methyl benzoate was added into the liquid products. Finally, a gas chromatograph (Agilent GC 6890N, Agilent Technologies Co., Ltd., Santa Clara, CA, USA) equipped with a flame ionization detector (FID) and a J&W 1701 (30.0 m × 250 µm × 0.25 µm) capillary column was used to quantitatively analyze the mixture solutions.
(2)$$Con_{C_{2}H_{2}\;}\left(\%\right)=\frac{n_{\mathrm{DMO}}+n_{\mathrm{DMM}}+n_{\mathrm{DMF}}}{n_{C_{2}H_{2}}}\times 100\%\;\;\;\;\;\;\;\;\;\;\;\;\;\;\;\;\;\;\;\;\;\;\;\;$$
(3)$$Sel_{\mathrm{DMO}}\;\left(\%\right)=\frac{n_{\mathrm{DMO}}}{n_{\mathrm{DMO}}+n_{\mathrm{DMM}}+n_{\mathrm{DMF}}}\times 100\%\;\;\;\;\;\;\;\;\;\;\;\;\;$$
(4)$$Sel_{\mathrm{DMM}}\;\left(\%\right)=\frac{n_{\mathrm{DMM}}}{n_{\mathrm{DMO}}+n_{\mathrm{DMM}}+n_{\mathrm{DMF}}}\times 100\%\;\;\;\;\;\;\;\;\;\;\;\;\;\;\;\;\;\;\;\;\;\;\;\;\;\;$$
(5)$$Sel_{\mathrm{DMF}\;}\left(\%\right)=\frac{n_{\mathrm{DMF}}}{n_{\mathrm{DMO}}+n_{\mathrm{DMM}}+n_{\mathrm{DMF}}}\times 100\%\;\;$$
(6)$$Y_{\mathrm{total}}\;\left(\%\right)=Con_{C_{2}H_{2}}\times \left(Sel_{\mathrm{DMM}}+Sel_{\mathrm{DMF}}\right)\times 100\%\;\;\;\;$$
where *Con* is the conversion of acetylene, and *n* represents moles of DMO, DMM, and DMF. *Sel* is the selectivity, and *Y*_total_ is the yield of unsaturated diesters.

## 3. Results and Discussion

### 3.1. Structure and Morphology of Nanocatalyst

TEM was used to characterize the micromorphologies of the nano-Pd and nano-Fe/Pd. Figure 1 shows the TEM images and average particle sizes of bimetal nanocatalysts. As seen in Figure 1c,d, the average particle size for nano-Pd and nano-Fe/Pd is (5.7 ± 0.78) nm and (3.7 ± 0.85) nm, respectively. The particle sizes of nano-Fe/Pd shrank, indicating that the introduction of Fe to the Pd lattice could slow down the crystallite growth of nanoparticles [15]. The TEM-EDX (Appendix A) of the nano-Fe/Pd nanoalloy catalyst further indicated that Pd and Fe were uniformly distributed in the alloy nanocatalyst. The particles morphologies and size of nanocatalysts revealed that, as expected, using PVP as a dispersant and NaBH_4_ as a reducing agent can obtain alloy nanoparticles with smaller particle sizes [19].

Figure 2 shows a high-resolution transmission electron microscopy (HRTEM) image of nano-Pd and nano-Fe/Pd. The lattice spacing of nano-Pd and nano-Fe/Pd is 0.230 nm and 0.228 nm, which corresponds to the separation of the Pd (111) lattice planes, respectively [20]. Meanwhile, the attached SAED pattern shows five sharp rings assigned to (111), (200), (220), (311), and (331) reflections of fcc Pd [20]. However, the lattice parameter of nano-Fe/Pd was higher than the lattice parameter (0.224 nm) of the FePd solid solution [21]. It is most likely that the Fe content in nano-Fe/Pd was lower and did not change the lattice spacing of Pd significantly.

To further elucidate the effect of Fe species on crystalline structures of the bimetallic catalyst, the crystalline structures of nanocatalysts were examined by XRD. Three diffraction peaks at 39.9°, 46.4°, and 67.7° of the monometallic nano-Pd XRD patterns (Figure 3) are attributed to Pd (111), Pd (200), and Pd (220) of the metallic Pd phase (JCPDS No. 46-1043), respectively. Meanwhile, PdO (111) and PdO (200) of the PdO phase (JCPDS No. 46-1211) are detected in nano-Pd particles [20,22]. It is worth noting that compared to the nano-Pd catalyst, the locations of Pd (111), Pd (200), and Pd (220) diffraction peaks of the bimetallic nano-Fe/Pd sample shifted slightly toward a higher angle. Furthermore, the diffraction peaks of Pd (111) of nano-Fe/Pd become broader and shorter compared to nano-Pd. Those phenomena have been construed as a sign of Fe-Pd alloy formation and that they help enhance catalyst stability [14,20]. Meanwhile, the peaks of PdO (111), PdO (200) crystal planes, and Fe species were not found in bimetallic samples. It is suggested that Fe atoms are partially incorporated into the palladium lattice, forming an Fe–Pd lattice solid solution, which could prevent palladium oxidation. XRD analysis shows that alloying Fe via an aqueous-based reduction procedure into the Pd crystalline host lattice probably caused the nano-Fe/Pd sample lattice contraction and the Pd d-band center shift, resulting in a stable and efficient catalyst for acetylene dicarbonylation.

XPS was also performed to analyze the chemical states of metallic species in the nanocatalyst to study their interaction. Appendix A depicts the Fe 2p XPS spectra of the bimetallic samples. It can be seen that the 2P_3/2_ level spectra of Fe are fitted with three peaks at 707.8 eV, 710.4 eV, and 712.6 eV, which are assigned to the species of metallic Fe^0^, Fe^2+^, and Fe^3+^, respectively [15,23,24]. The Fe^2+^ and Fe^3+^ species of the catalyst surface are most likely oxidized by Fe^0^ species in the air. Figure 4 shows the 3d_5/2_ level spectra of Pd in nanoalloy catalysts are better fitted with three symmetrical peaks at approximately 334.7, 335.8, and 337.5 eV, assigned to metallic Pd, PdO, and Pd^δ+^ (2 ˂ δ ≤ 4) species, respectively [25,26]. To further evaluate the expected composition and elemental mapping of the catalyst samples, XPS and ICP-OES elemental analyses were used. Interestingly, XPS results show that the molar ratio of Pd to Fe in the nano-Fe/Pd catalyst is 6.2:1, while ICP-OES analysis results are relatively close to the nominal atomic ratios (*n*_Pd_:*n*_Fe_ = 9.8:1). This proves Fe elements tend to enrich at the surface, rather than migrate to the core of the nano-Fe/Pd.

Table 1 summarizes all of the binding energies of Pd species and their atomic concentration in the catalyst achieved from fitting the experimental XPS peaks. Therein, the molar ratio of Pd^0^/Pd^2+^ in the monometallic sample (0.90) was much lower than that of the bimetallic catalyst (1.24). This result was due to Fe atom preferences losing their electrons to Pd atoms in bimetallic catalysts, which led to Pd in the metallic form to construct the crystal architecture [26]. It is reported that the change in binding energies of Pd species and their atomic concentration in bimetallic catalysts could promote the palladium lattice rearrangement and develop favorable sites for improving catalyst activity and catalytic stability [27].

H_2_-TPR was implemented to investigate the mutual effect of the Pd and Fe species in the catalyst. The negative peak at approximately 0 °C in the H_2_-TPR profile (Figure 5) of the nano-Pd catalyst is attributed to the hydrogen release from palladium hydride (PdHx) decomposition [15,26]. The peak at 72 °C is ascribed to the reduction of PdO to metallic Pd (Figure 4a). As is shown in Figure 5b, the reduction peak of PdO to the metallic Pd of nano-Fe/Pd catalysts appeared at 66 °C. The nano-Fe/Pd catalyst also has a broad peak, which is associated with the reduced peaks of Fe_2_O_3_ to Fe_3_O_4_ (305 °C) and Fe_3_O_4_ to FeO (430 °C). It is worth noting that the reduced peak of the corresponding metal oxides was lower than the experimental results from the literature [26]. It was proposed that the interaction between two metal species affected the reducibility of the catalyst. Meanwhile, the H_2_ consumption (Appendix A) of catalysts was calculated by integrating the corresponding peak areas with that of the standard sample (AgO), which further proves that alloying a small amount of Fe into the nano-Pd crystalline might increase the electron density around the Pd atom, which could improve catalyst activity and stability for acetylene dicarbonylation.

### 3.2. Catalyst Performance and Evaluation

Based on the previous studies, KI and Brønsted acids are an essential promoter in acetylene dicarbonylation [2,3,4,6,7,8]. However, the Brønsted acids are very corrosive, hence, we used KI as the co-catalyst of acetylene dicarbonylation. The bimetal nanoalloy catalysts (marked as M/Pd; M = Fe, Co, and Cu) were prepared by aqueous-based reduction methods to study their catalytic performance for the acetylene dicarbonylation. The XRD patterns and TEM images of nano-Cu/Pd and nano-Cu/Pd are given Appendix A. As shown in Table 2, the monometallic catalyst nano-Pd received a relatively low conversion (66.8%) and a high selectivity (99.5%) for the acetylene substrate and unsaturated 1, 4-dicarboxylic acid diesters DMF and DMM, respectively. When Co atoms are incorporated into the Pd nanoparticles by simultaneous reduction, they inhibit nano-M/Pd catalyst activity. In contrast, when Fe and Cu are alloyed into the nanocatalysts, the catalytic activity and selectivity of the nanoparticles are significantly improved. 

These results suggest that introducing Fe and Cu elements to palladium catalysts significantly promotes their activity and selectivity for alkyne dicarbonylation. For example, the nano-Fe/Pd catalyst (Table 1, entry 2) achieves a high dicarbonylation selectivity of DMF and DMM (99.7%) as well as good acetylene conversion (83.9%). Meanwhile, we observed many black solids (palladium black) being generated in the case of PdCl_2_ [28]. It was suggested that even the PdCl_2_/KI system (Table 1, entry 5) has good conversion and selectivity, however, this homogeneous catalyst system is not an ideal option for acetylene dicarbonylation. Remarkably, during catalyst evaluation, we found that the liquid product for nano-Pd and nano-Cu/Pd exhibits a dark brown and a reddish brown for nano-Fe/Pd systems. It is widely known that metal leaching is a crucial obstacle for studying CO-containing heterogeneous catalyst systems [29,30]. According to previous research, the color of the reaction liquid depends on I_2_, which is generated from the I^−^ anion by the air as the ultimate oxidant [2,6,31].

To prove the color from I_2_, a certain amount of K_2_S_2_O_3_ was added into the product supernatant. As Appendix A shows, the color of the nano-Pd and nano-Cu/Pd product supernatant underwent a noticeable change from dark-brown to pale yellow after being treated by K_2_S_2_O_3_. In contrast, the color of the nano-Fe/Pd supernatant transformed from brown to colorless. Therefore, ICP analysis was used to determine the Pd metal content in the supernatant of each system to reveal the origin of the different colors of the supernatant. It is worth noting that ICP results proved Pd significant leaching in the nano-Pd and nano-Cu/Pd systems accounted for approximately 34.2% of the total Pd and only 1.3% of the total Pd in the nano-Fe/Pd catalyst. Screening of the catalyst shows that Nano-Fe/Pd is an excellent catalyst in acetylene dicarbonylation.

According to previous reports [2], the acetylene conversion is shown to be strongly influenced by the temperature, CO pressure, reaction duration, and co-catalyst KI dosage. Figure 6 shows the temperature effect on the catalytic performance of the nano-Fe/Pd catalyst for acetylene diycarbonylation. It was discovered that the reaction temperature range from 30 to 50 °C significantly boosted the product yield of DMM and DMF from 9.2% to 92.3%. The overall output decreased from 92.3 to 89.8 °C at 70 °C as soon as the temperature exceeded 50 °C. The total yield demonstrated the volcano-type pattern by increasing initially, then declining as the reaction temperature rose. As previously reported [6], rather than the catalyst deactivation, the kinetic effect is what causes the conversion of acetylene to decrease when the reaction temperature rises. Through the experiment, the optimal reaction temperature for acetylene diycarbonylation is 50 °C, the product yield (DMM and DMF) could acquire a high yield of 92.3%, and the ratio of cis-to-trans isomer reached 3.3:1.6.

Figure 7 illustrates the effect of CO pressure on yield and the cis–trans isomer ratio of acetylene dicarbonylation. It was found that when the pressure of CO was 0.8 MPa, the total yield was only 62.5%. When the pressure of CO increased to 1.2 MPa, the total yield improved to 86.5%. In addition, the ratio of cis-to-trans isomers is essentially constant (2.6:1) within the range from 0.8–1.2 MPa of the partial pressure of CO. When the CO pressure was 1.8 MPa, the total yield was 92.3%. However, when the partial pressure of CO was increased to 2.0 MPa, the total yield fell to 81.3%, and the ratio of cis-to-trans isomers increased from 2.6:1 to 3.6:1. This result could be attributed to increased CO pressure causing palladium species coordination change, which is beneficial for trans isomer production. 

Figure 8 shows the reaction time effect on the catalytic activity of a nano-Fe/Pd catalyst. It was found that the total product yield (DMF and DMM) increases with the reaction time and reaches 96.3% after 8 h. Meanwhile, with the reaction time extended, the cis–trans isomer ratio remained constant at approximately 3:1 after 6 h. Under different reaction time conditions, the acetylene was almost consumed after more than 8 h.

Additionally, as shown in Figure 9, the reactivity gradually rises as KI dosage increases. When the amounts of KI increases to 80 mg, the acetylene conversion rate reaches 98.4% and the reaction shows a high selectivity of unsaturated 1,4-dicarboxylic acid esters up to 99.7%. Meanwhile, as the amount of KI further increases, the cis–trans isomer ratio decreases. It is mainly because of the I^-^ coordination configuration of intermediate products, which is conducive to the trans-insertion of coordinated acetylene. All the nano-Fe/Pd catalysts exhibited higher activity than the other reported catalysts Pd(nanosheet)/AC [6], Pd/Fe_2_O_3_ [7], and homogeneous PdCl_2_ system [28].

Similarly, the stability of the nano-Fe/Pd catalyst was further tested by a cycling experiment. The catalyst was retrieved by centrifugal washing with ethyl acetate and ethanol followed by freeze-drying. As shown in Figure 10, nano-Fe/Pd catalysts being recycled and reused for up to five consecutive cycles showed high activity for acetylene dicarbonylation reactions. However, the yield of unsaturated esters slightly decreased with the increase in the cyclic number and showed 84.4% in the fifth cycle. TEM-EDS (Appendix A) analysis revealed that Fe and Pd were uniformly distributed in the spent catalysts, the amount of Pd in the fifth recycled catalyst was approximately 69.8 wt% (fresh catalyst 61.2 wt%, Appendix A) by ICP analysis. It indicates that metal leaching was not the main reason for the loss of activity. Additionally, the XPS results of the C 1s of recovered catalysts show the formation of carbon deposition (288.6 eV, Appendix A). The coke was also observed on the recycle catalyst surfaces by a TEM image (Appendix A). As a result, we assumed that coke deposition on the catalyst surfaces and blocking of the active sites were the main reasons for the catalyst activity loss. 

It was reported that I^-^ was a weak base ligand that plays a pivotal role in the acetylene dicarbonylation reaction activity and mechanism [2,31]. Hence, the effect of the cocatalyst on the 1,4-dicarboxylic acid esters was also examined. As seen in Figure 11, replacing KI with NaCl and KCl as cocatalysts resulted in almost no reaction, indicating that the reaction hardly ever proceeds without iodine additives. In contrast, I_2_ was used as a cocatalyst and obtained a good yield of 56.3%. It could be due to hydriodic acid being continually generated in situ during the reaction process, with the latter initiating the catalytic cycle by forming a Pd-I species. Overall, it was believed that the cocatalyst KI was the most effective co-catalyst available. 

### 3.3. Mechanism Studies

According to reports in the literature, CO adsorption IR-spectrum on a Pd particle surface has three types of stretching bands: gaseous CO (2200~2100 cm^−1^), linearly adsorbed CO on the metallic Pd^0^ species (2100~2000 cm^−1^), bridged adsorbed CO on Pd^0^ (2000~1900 cm^−1^), and three-fold hollow bridge sites (1900 cm^−1^~1800 cm^−1^) [21,32,33]. However, IR spectroscopic studies of Pd nanoparticles, which are synthesized by reducing PdC1_2_ and stabilized with PVP, produce abnormal effects in the inverting of the IR band direction of adsorbed CO species [34,35]. Thus, the CO molecular probe infrared spectroscopy only was used to study surface sites for CO adsorption after Fe atoms were alloyed into Pd nanoparticles.

Figure 12 shows the FT-IR spectra of adsorbed CO on nanocatalysts. However, no obvious characteristic peaks of CO adsorption IR-spectra for the nano-Pd catalyst were detected. In contrast, the relatively strong characteristic peaks of CO adsorption IR-spectra are found for the nano-Fe/Pd catalyst. It has been discovered that linear CO adsorbed on highly dispersed metallic Pd has a higher intensity of the IR peaks at 2085 and 2060 cm^−1^. The weak shoulder peak at 1993 cm^−1^ is attributed to CO adsorbed on Pd (100) bridge sites. The considerably broad and weakening peaks from 1890 to 1800 cm^−1^ are ascribed mainly to the three-fold hollow and bridge sites of CO molecules adsorbed on planes. Meanwhile, two distinct characteristic peaks located at 2172 and 2123 cm^−1^ are signals corresponding to the gaseous and weakly adsorbed CO. To verify that the gas phase or adsorbed species on the surface of nano-Fe/Pd are responsible for those bands. After using He sweeps for 15 min, the bands were still visible. It was reported that the peaks at 2175 cm^−^^1^ and 2118 cm^−^^1^ can be attributed to linearly adsorbed CO on Pd^2+^ and Pd^+^, respectively [36]. Thus, we believe that the characteristic peaks appeared at 2175 cm^−1^ and 2123 cm^−1^ of linear CO adsorption on Pd^2+^, Fe^δ+^, and Pd^+^ on the surface of nano-Fe/Pd, which was the newly emerged nano-Fe/Pd catalyst [21,37]. The results of the CO FT-IR further provided that the added low-content Fe into the Pd nanoparticles can synergistically generate a favorable site for facile CO adsorption on the nano-Fe/Pd catalyst surfaces.

To gain further insight into the synergy effects between Pd and Fe, a theoretical calculation for this system was performed. First, we examined the pure Pd (111) surface to evaluate the most favorable way for to achieve CO and C_2_H_2_ adsorption, followed by a comparison of the changes on the surface alloy systems (Appendix A). The calculation result shows that the hollow sites (fcc) on the pure Pd (111) surface are more conducive to CO adsorption (*E*_ads(CO)_= −200.7 KJ/mol). However, after replacing one Pd in the topmost layer of Pd (111) with the Fe atom, the top site of the Fe atom of the Fe/Pd (111) (Figure 13b) was found to be the most favorable site for CO adsorption (*E*_ads(CO)_= −52.3 KJ/mol).

Table 3 shows the adsorption energies and bond lengths of C–O of CO and C≡C of C_2_H_2_ of the adsorption state on the pure Pd (111) and Fe/ Pd (111) models, respectively. Remarkably, the CO adsorption energy *E*_ads_ on the Pd (111) model surface ranged from −139.4 kJ/mol to −200.7 kJ/mol, which was roughly 150 kJ/mol less than CO adsorption on Fe/Pd (111) model surfaces at corresponding sites. It is suggested that the alloying of Fe atoms into the palladium lattice weakens the adsorption capacity of carbon monoxide of the catalyst and could help to suppress formed Pd carbonyls, without resulting in active metal component leaching. According to previous studies, C≡C bond activation is the key step in acetylene carbonylation. Therefore, the C≡C bond length change of adsorbate C_2_H_2_ on the pure Pd (111) and Fe/Pd (111) model surfaces were compared. Table 3 shows that the C≡C bond length is 1.37 Å, which is approximately 0.16 Å longer than the free triple-bond of the C_2_H_2_ molecule (1.21 Å) on pure Pd (111) fcc adsorption sites. Moreover, for adsorbate C_2_H_2_ on the Fe/ Pd (111) surface, the C≡C bond length is 1.41Å, which is longer than that on pure Pd (111) at 0.04 Å. 

Furthermore, Bader charge analysis demonstrated that Fe in an alloy Fe/Pd (111) model tends to lose its electron (0.65 a.u.) to the Pd atom, which is consistent with the previous XPS analysis [38]. Combined with XPS, FT-IR, and DFT calculations, these results illustrate that alloying a small amount of Fe into the sub-nano-Pd shell could shrink the catalytic nanoparticle size and, thus, increase the electron density around Pd-adjacent Fe. More significantly, DFT calculations showed that the adsorption of C_2_H_2_ on the Fe/Pd (111) surface was still very strong (106.9 KJ/mol). The Bader charge analysis (Figure 14) revealed that C_2_H_2_ adsorbed on Pd (111) only has a −0.24 a.u. electron. However, when the C_2_H_2_ adsorbed on Fe/Pd (111), the negative charge increased to 0.4 a.u. It was indicated that the nano-Fe/Pd catalyst donates more electrons to C_2_H_2_ located on the surface. DFT calculations revealed that CO carried a negative charge (−0.22 a.u.) adsorption on the nano-Pd surface. In contrast, nano-Fe/Pd donated more electrons to CO 2π* orbitals, thus, enhancing the negative charge of CO (−0.30 a.u.), weakening the C–O adsorption strength, and changing the CO adsorption sites. It was believed that the electron-rich nature of nano-Fe/Pd would alter the adsorption behavior of CO and C_2_H_2_, thus, improving the catalytic performance of the nano-Fe/Pd catalyst. Based on these results, we expected that the electron-rich Pd center on the surfaces of the catalysts would play a significant role in the acetylene dicarbonylation reaction [39]. In comparison to the nano-Pd catalyst, alloying Fe atoms into the Pd lattice allows for the formation of more Pd active species capable of activating the carbon–carbon triple bond of acetylene. Furthermore, the Fe atoms on the catalyst surfaces may aid in CO migration and activation. Overall, alloying Fe into the nano-Pd catalyst improved the stability of the nano-Fe/Pd catalyst as well as effectively improving catalyst activity of the Pd-based catalyst in the acetylene dicarbonylation reaction.

## 4. Conclusions

In summary, a stable acetylene dicarbonylation bimetallic catalyst has been successfully prepared by alloying iron into the palladium lattice. Nano-Fe/Pd not only exhibits excellent activity (98.4%) and selectivity (99.0%) under the optimum conditions but also effectively restrains metal Pd leaching under a high pressure CO atmosphere. Detailed electron microscopy and spectroscopy characterization revealed that Fe atoms were incorporated into the palladium lattice and formed an Fe/Pd alloy. Moreover, the electrons of Fe naturally transfer to Pd in an alloy, resulting in an electron-rich Pd center, which significantly influences the electronic structure and morphology of the nano-Fe/Pd catalyst. Theoretical calculations further indicated that alloying iron into the palladium lattice significantly weakened the interaction strength of CO adsorbed on the nano-Fe/Pd surface and enhanced the activation of the C≡C bond of C_2_H_2_. The strong synergistic effects between the two metallic elements, which are expected to improve activity, also suppress active metal component leaching in acetylene dicarbonylation.

## Figures and Tables

**Figure 1 nanomaterials-12-03803-f001:**
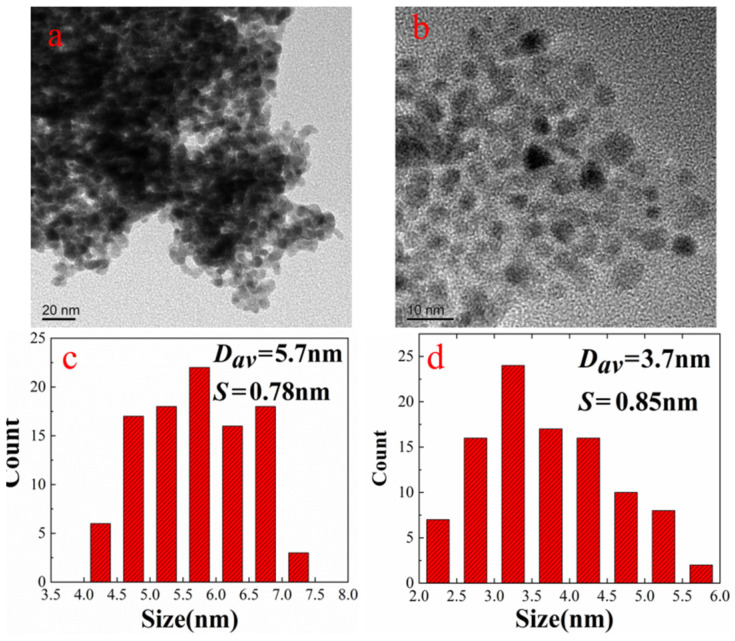
TEM images of nano-Pd (**a**) and nano-Fe/Pd (**b**) nanoparticles and corresponding particle size distribution (**c**,**d**).

**Figure 2 nanomaterials-12-03803-f002:**
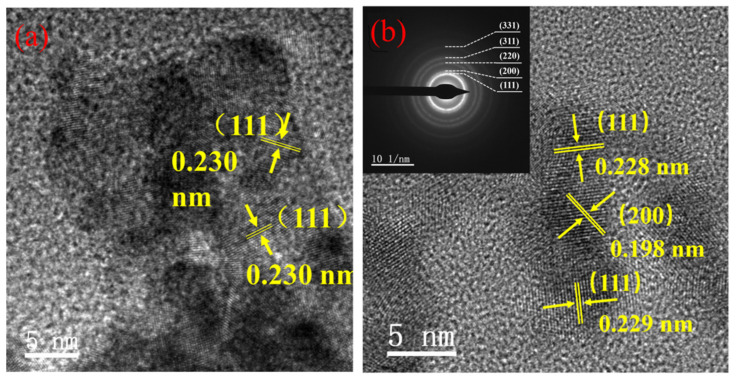
(**a**) HRTEM image of nano-Pd and (**b**) nano-Fe/Pd (Inset shows the corresponding SAED pattern).

**Figure 3 nanomaterials-12-03803-f003:**
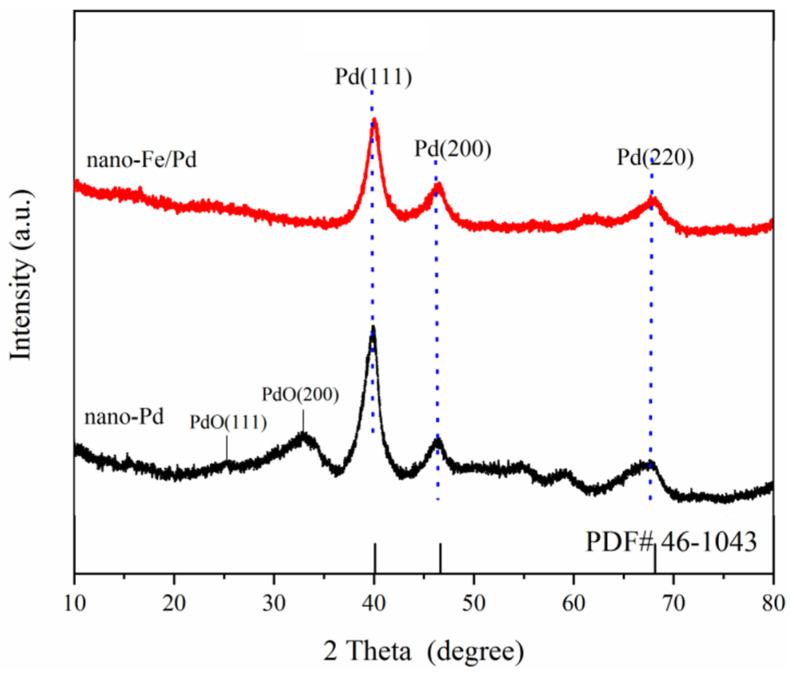
XRD patterns of nano-Fe/Pd and nano-Pd samples.

**Figure 4 nanomaterials-12-03803-f004:**
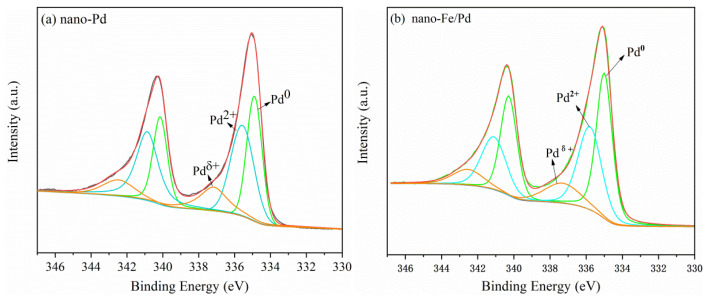
High resolution Pd 3d XPS spectra of nano-Pd (**a**) and nano-Fe/Pd (**b**).

**Figure 5 nanomaterials-12-03803-f005:**
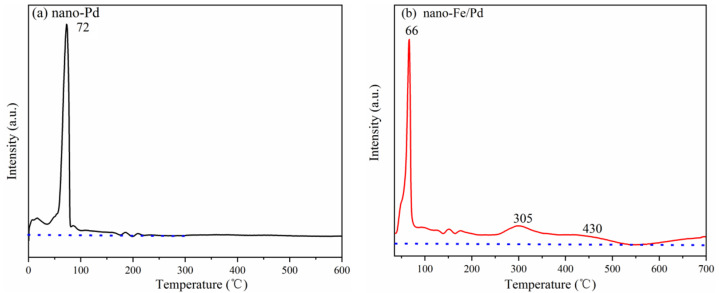
H_2_-TPR spectrum of nano-Fe/Pd (**a**) and nano-Pd (**b**) catalyst.

**Figure 6 nanomaterials-12-03803-f006:**
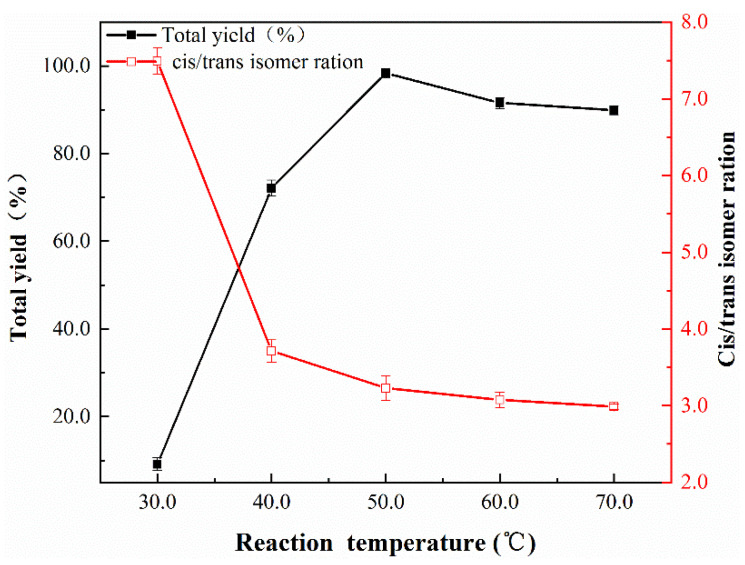
Effect of temperature effect on nano-Fe/Pd catalyst activity in the reaction. Reaction conditions: catalyst (10 mg), acetylene (11 mmol), MeOH (20 mL), KI (112 mg), P_CO_ = 1.8 MPa, P_total_ = 4.0 MPa, t = 6 h.

**Figure 7 nanomaterials-12-03803-f007:**
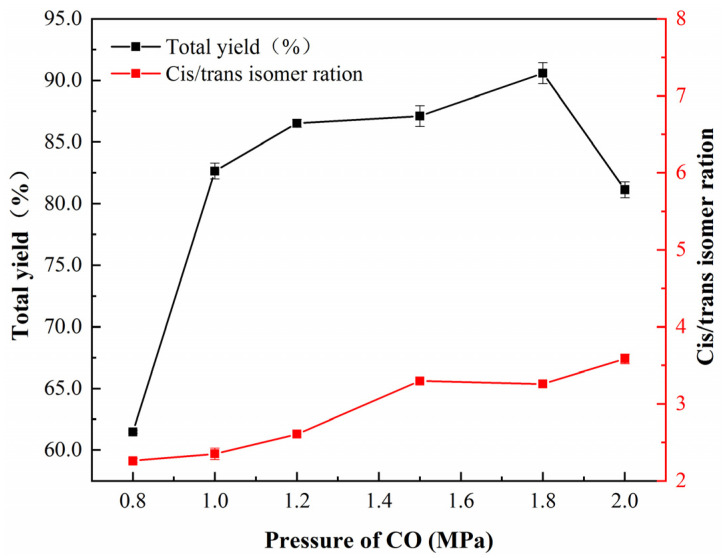
Effect of CO partial pressure on nano-Fe/Pd catalyst activity in the reaction. Reaction conditions: catalyst (10 mg), acetylene (11 mmol), MeOH (20 mL), KI (112 mg), P_total_ = 4.0 MPa, T = 323.2 K, t = 6 h.

**Figure 8 nanomaterials-12-03803-f008:**
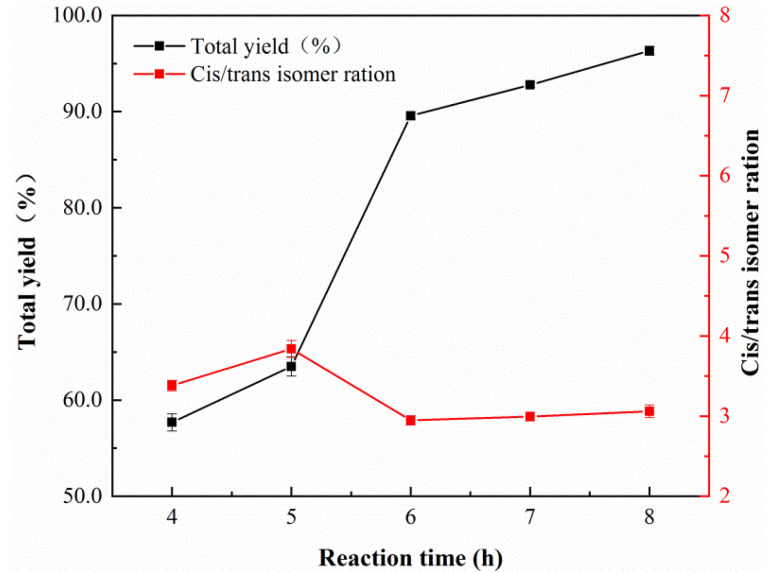
Effect of reaction time on nano-Fe/Pd catalytic activity in the reaction. Reaction conditions: catalyst (10 mg), acetylene (11 mmol), MeOH (20 mL), KI (112 mg), P_CO_ = 1.8 MPa, P_total_ = 4.0 MPa, T = 323.2 K.

**Figure 9 nanomaterials-12-03803-f009:**
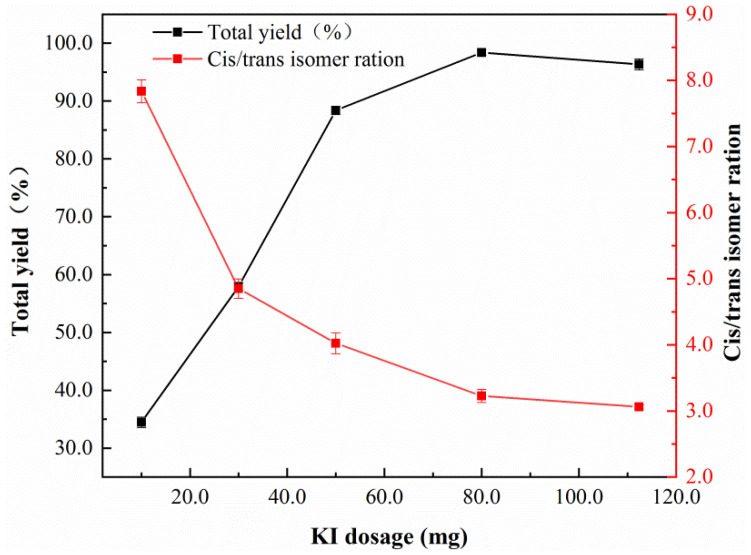
Effects of KI dosage on the nano-Fe/Pd catalyst activity in the reaction. Reaction conditions: catalyst (10 mg), acetylene (11 mmol), MeOH (20 mL), P_CO_ = 1.8 MPa, P_total_ = 4.0 MPa, T = 323.2 K, t = 8 h.

**Figure 10 nanomaterials-12-03803-f010:**
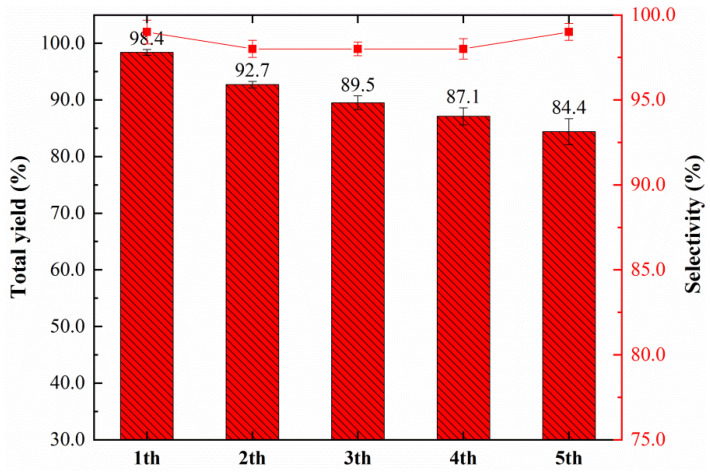
Recycle study of nano-Fe/Pd catalyst for acetylene dicarbonylation and selectivity of production (red line). Reaction conditions: catalyst (10 mg), acetylene (11 mmol), MeOH (20 mL), KI (80 mg), PCO = 1.8 MPa, P_total_ = 4.0 MPa, T = 323.2 K, t = 8 h.

**Figure 11 nanomaterials-12-03803-f011:**
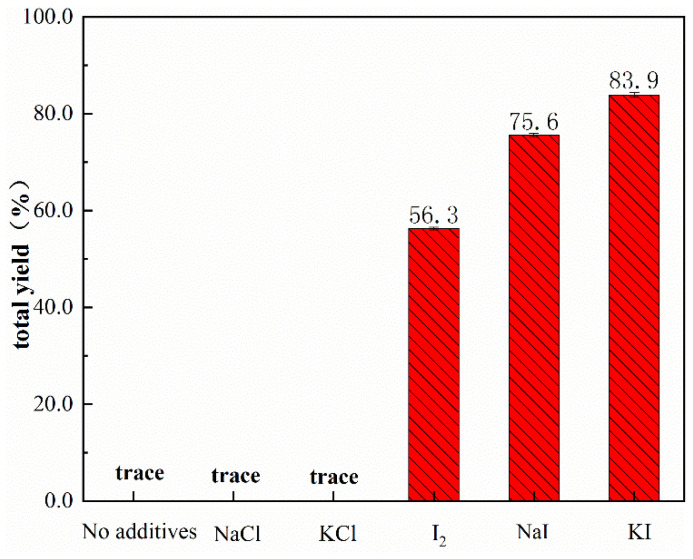
Effect of different cocatalysts on the dicarbonylation of acetylene. Reaction conditions: catalyst (10 mg), acetylene (11 mmol), MeOH (20 mL), KI (112 mg), P_CO_ = 1.8 MPa, Ptotal = 4.0 MPa, T = 343.2 K, t = 5 h.

**Figure 12 nanomaterials-12-03803-f012:**
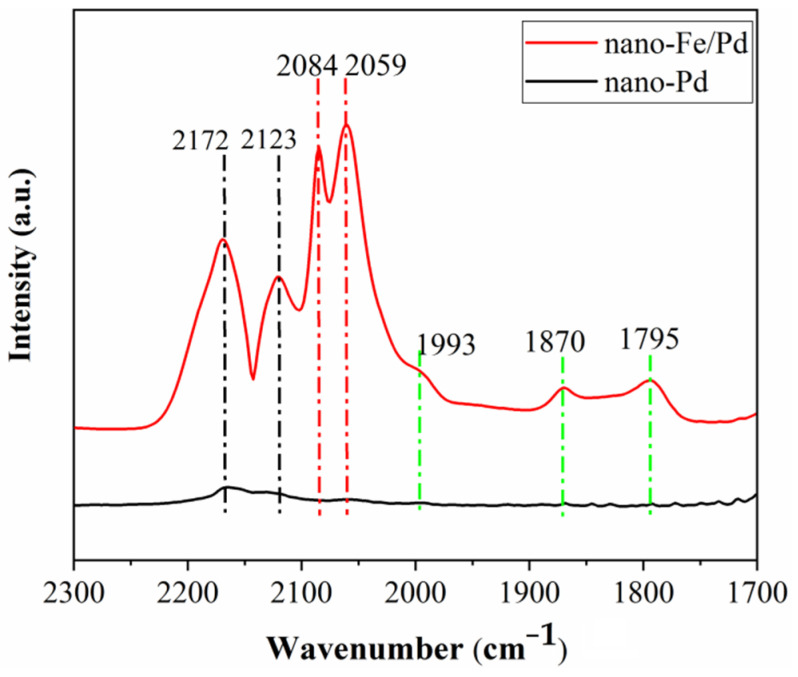
IR spectra of CO adsorbed on nanocatalyst.

**Figure 13 nanomaterials-12-03803-f013:**
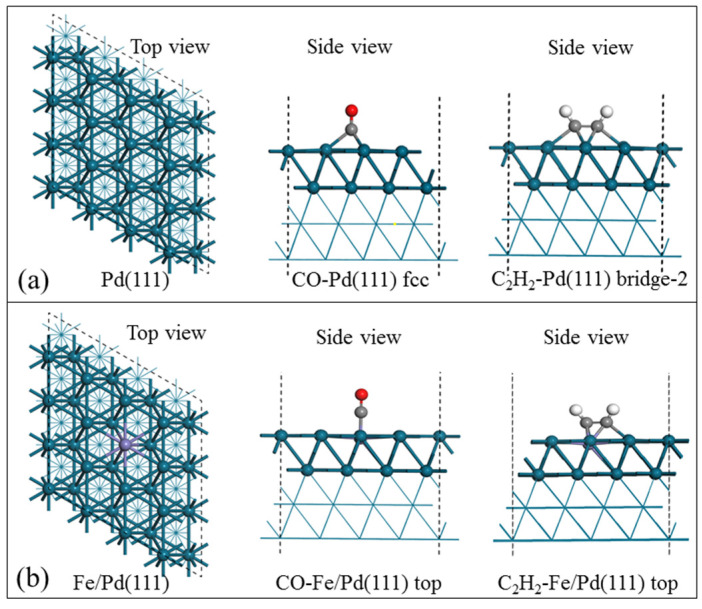
DFT optimized structures. (**a**) Pd (111) and (**b**) Fe/Pd (111), and CO or C_2_H_2_ energetically favored adsorption configurations on catalyst surfaces.

**Figure 14 nanomaterials-12-03803-f014:**
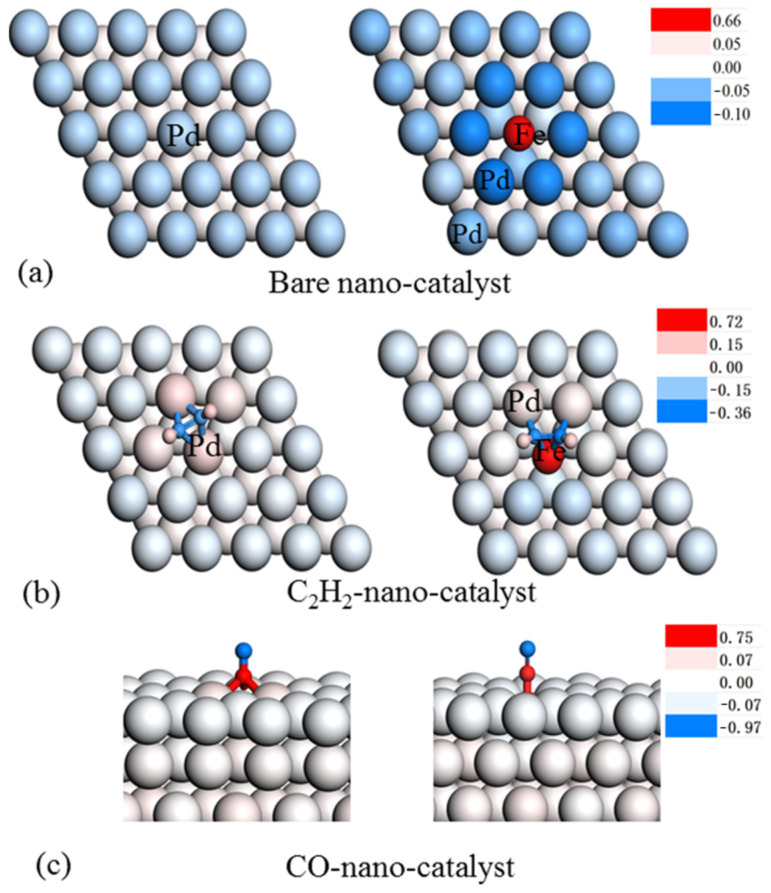
Bader charge analysis of bare nano-catalysts (**a**), C_2_H_2_ adsorbed on the nano-catalyst surface (**b**) and CO adsorbed on the nano-catalyst surface (**c**). The colors indicate the Bader charge of each atom. The profile of each atom is shown on the right.

**Table 1 nanomaterials-12-03803-t001:** Relative atomic concentration (%) of Pd in the samples by XPS analysis.

Sample	Nano-Pd		Nano-Fe/Pd	
	Binding Energy (eV)	Atom%	Binding Energy (eV)	Atom%
Pd^0^	334.9	38.67	334.9	47.17
Pd^2+^	335.6	43.14	335.8	38.00
Pd^δ+^	337.2	18.2	337.5	14.82

**Table 2 nanomaterials-12-03803-t002:** Activity comparison of different catalysts for acetylene dicarbonylation.

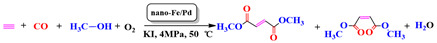
Entry	Catalyst	*Sel* (%)	Total Yield (%)
DMO	DMF	DMM
1	Pd	0.5	29.0	70.5	66.8
2	Fe/Pd	0.3	26.2	73.5	83.9
3	Co/Pd	0.3	17.3	83.4	10.4
4	Cu/Pd	0.1	28.2	71.7	79.2
5	PdCl_2_	0.1	26.6	73.3	82.8

Reaction conditions: catalyst (10 mg), acetylene (11 mmol), MeOH (20 mL), KI (112 mg), P_CO_ = 1.8 MPa, P_air_ = 2.2 MPa, P_total_ = 4.0 MPa, T = 343.2 K, t = 5 h. The conversion, the yield, and selectivity were determined by GC using methyl benzoate as the internal standard.

**Table 3 nanomaterials-12-03803-t003:** The Calculated Binding Energies (*E*_ads_) of CO and C_2_H_2_ on the surface of the sample, as well as the bond length C≡O of CO and C≡C bond of C_2_H_2_ of adsorbate-containing sample.

	Adsorption Sites	*E*_ads_ (KJ/mol)		Bond-Length (Å)	
		*E*_ads_ (CO)	*E*_ads_ (C_2_H_2_)	C≡O Bond	C≡C Bond
	p (4 × 4) Pd (111)				
1	hcp	−199.1	−185.2	1.19	1.37
2	fcc	−200.7	−186.6	1.19	1.37
3	top	−139.4	−77.6	1.16	1.26
4	Bridge	−182.4	−152.4	1.17	1.32
5	Bridge-2	-	−191.2	-	1.39
	p (4 × 4) Fe/Pd (111)				
6	hcp	−43.5	−77.4	1.20	1.39
7	fcc	−23.9	−102.9	1.18	1.41
8	top	−52.3	−106.9	1.17	1.41
9	bridge	−50.0	−78.7	1.90	1.39
10	Bridge-2	-	−103.2	-	1.41

## Data Availability

The data is available upon reasonable request from the corresponding author.

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
