# Peer review of "Alloying Iron into Palladium Nanoparticles for an Efficient Catalyst in Acetylene Dicarbonylation"

_nanomaterials, 2022, doi:10.3390/nano12213803_

Round 1

Reviewer 1 Report

The paper deals with alkyne carbonylation using bimetallic metal nanoparticles. In their work the authors try to address Pd leaching by alloying palladium with small amunts of iron arguing that their synthesis method results in FePd alloys altering the electron density of Pd after alloyng and try to provide an explanation of the encountered catalytic behaviour on the basis of DFT calculations.

The authors claim that the have synthesised a FePd alloy and reasonable doubts may be raised on this afirmation or that Fe tunes the electronic-geometrical structure of the catalysts. A modification of the DOS of a FePd alloy may be assumed but the geometrical structure of the FePd sites is far from being addressed with the described experiments. The authors claim that charge transfer from iron to palladium weakens CO adsorption capacity  and enhances C-C activation. The problem with this is that the authors use DFT to model just one Fe atom dispersed in Pd(111) surface and a FCC nanaoparticle of the reported size has a considerable proportion of (100) faces and edge atoms that has a different reactivity, therefore providing a very simplified view of the catalytic process.

Comments above are the general comments on the paper content but more detailed comments  follows:

Section 2.5

In this section the use of KI that is further discussed must be described as well as the procedure for cycling the catalysts

Page 4 lines 147-151

The authors state: "Moreover, the TEM-EDX (Figure S1) of the nano-Fe/Pd
nanoalloy catalyst further indicated that Pd and Fe were uniformity distributed in alloy nanocatalyst. The particles morphologies and size of nanocatalysts revealed that, as expected, using PVP as a dispersant and NaBH4 as a reducing agent can obtain anlloy nanoparticles with smaller particle sizes
"

There is no problem is assuming the uniform distribution of both metals but this nos necessarily means that they have obtained an alloy. Iron atoms may be at the Pd surface. According to the Fe-Pd binary phase diagram for a ~10%Fe-90%Pd alloy the thermodynamic equilibrium suggests a mixture  of FePd3 and FePd solid solution that may present order-disorder phenomena since the small particle size.

High resolution electron microscopy with SAED and EDS analysis should clarify this point and clearly demonstrate the geometrical structure of the synthesised system

Moreover, a rapid view of the EDS data in the supplementary material evidences sgnals at ~1,0 and ~2,5 keV that may be ascribed to Na and Cl which may be present in the solid as a result of the synthesis method, procedures for eliminating the contamination are not described in the experimental section or through the text. Therefore, is there boron, sodium or chlorine on the catalyst surface? This must be addressed since it could play a key role in the catalytic properties.

Page 5.

"Three diffraction peaks at 39.9°, 46.4°, and 67.7° of the monometallic nano-Pd XRD patterns (Figure 2) attributed to Pd (111), Pd (200), and Pd (220) of metallic Pd phase (PDF 46-1043). ......... It is worth noting that compared to the nano-Pd catalyst, the locations of Pd (111), Pd (200), and Pd (220) diffraction peaks of the bimetallic nano-Fe/Pd sample shifted slightly toward a higher angle."

The shift towards higher angles may be due to the formation of the solid solution but assuming Vegard's law is fulfilled a diffraction angle shift of 0,3-0,4º should be expected for the (111) diffraction line and up to 0,6-0,8 for the (220) one. This cannot be observed in figure 2 where such shifts are not observed.

Therefore, the authors atatement ".... Fe atoms be partially incorporated into the palladium lattice and form an FePd lattice solid solution...." is at least doubtful.

The XPS lineshape is usually affected by multiple chemical environments,  vibrational excitations, and secondary electronic excitations (O'Connor et al. J. Phys. Chem. C 2021, 125, 19, 10685–10692), these results, particularly for metals, in asymmetric line shapes. Typically, Pd lineshape is best fitted with an asymmetric lorentzian shape. Thus, the deconvolution performed is highly hypothetical as it is in the c ase of iron where the signal-to-noise ratio does not allow an eitgh gaussian fitting,

Quantification of the TPR profile is mandatory to really ascribe the different events as the authors do (page 7)

Section 3.2

It is not clear the reasons why the authors introduce such a battery of systems without characteerization, composition or any kind of information apart from the activity and selectivity data. If they wish to extent the analysis to the whole set of systems they must proceed to characterise them.

In any case, they assign the differences to the formation of alloys but they forget to consider the presence of sodium (observed in the EDS) or that of chlorine which is surprising since PdCl2 (last entry in table 2) is roughly as active and selective as the nanoparticle catalysts.

Error bars must be provided for all cis/trans ratios and activity data to better define the tendencies (figures 5-9).

Seciton 3.3

The stretching mode of the CO molecule is symmetry forbidden and should appear at 2143 cm-1, exactly at the minimum between the two (highly smoothed "bands" at 2172 and 2123 cm-1. These two bands are the envelops of the rotational gas phase spectrum of the CO molecule. Therefore, if these bands do not appear in the spectrum is either becuase there is no CO in the IR cell or an experimental error occurred.

These experiments, if relevant, must be carried out in adequate cnditions to see if effectively there is an electronic interaction between Fe and pd that should be reflected in a shift in the position of the linear CO bands.

To publish this paper all these must be fixed and the additional experiments suggested performed.

Reviewer 2 Report

Yuchen Zhang et al. report novel palladium nanoparticles alloyed with iron. These nanoparticles are found to provide efficient catalytic activity in acetylene dicarbonylation, i.e., 98.4% yield and 99.7% selectivity for the unsaturated 1,4-dicarboxylic acid diesters. It was found that proves Fe ions enrich at the surface, not the core of the nano-Fe/Pd. Both experimental analysis and ab initio theoretical calculations are reported in the manuscript. Vienna ab initio simulation package (VASP) was used. The section 2 provides most details for experiments and calculations.

The paper is well structured and written. The results are consistently presented and support the conclusions. There some minor issues to be checked before publication:

1. In Figure 3b, the violet curve needs explanations and caption.

2. Lines 106-121 do not contain details on optimization of the slabs.

3. Is it possible to discuss a dependence of the 305 °C peak on the Fe concentration?

4. Reaction temperature has a peak at 50 °C, as it follows from Figure 5. However, the nontrivial behavior of the curve may suggest a shift of the peak to 45 °C for the corresponding temperature steps.

The manuscript can be accepted to Nanomaterials.

Round 2

Reviewer 1 Report

The authors has rteralised some minor modification in their manuscript but they have not addressed most of the previous comments

I am including along with their comments a detailed description of the previous considerations and the reasons the authors have not fulfilled them

Reviewer 3 Report

1. I recommend the authors to make corrections at the highlighted places in the attached file.

2. The authors should explain the mechanism of Acetylene Dicarbonylation on their Fe/Pd bimetallic catalyst surface.
